# Circumferential Strain as a Marker of Vessel Reactivity in Patients with Intradialytic Hypotension

**DOI:** 10.3390/medicina59010102

**Published:** 2023-01-02

**Authors:** Maciej Goździk, Sergiusz Ustyniak, Anna Zawiasa-Bryszewska, Agnieszka Płuciennik, Maja Nowicka, Magdalena Kaczmarska, Ludomir Stefańczyk, Ilona Kurnatowska

**Affiliations:** 1Department of Internal Medicine and Transplant Nephrology, Chair of Pulmonology, Rheumatology and Clinical Immunology, Medical University of Łódź, 90-153 Łódź, Poland; 2Department of Radiology and Diagnostic Imaging, Medical University of Łódź, 90-153 Łódź, Poland

**Keywords:** intradialytic hypotension, circumferential strain, vessel reactivity

## Abstract

*Background and Objectives*: Intradialytic hypotension (IDH) complicates 4 to 39.9% of hemodialysis (HD) sessions. Vessels’ reactivity disturbances may be responsible for this complication. Two-dimensional speckle tracking is used to assess arterial circumferential strain (CS) as a marker of the effectiveness of the cardiovascular response to the reduction of circulating plasma. *Materials and Methods*: The common carotid artery (CCA) and common iliac artery (CIA) CSs were recorded using ultrasonography in 68 chronically dialyzed patients before and after one HD session. *Results*: In patients with IDH episodes (*n* = 26), the CCA-CS was significantly lower both before (6.28 ± 2.34 vs. 4.63 ± 1.74 *p* = 0.003) and after HD (5.00 (3.53–6.78) vs. 3.79 ± 1.47 *p* = 0.010) than it was in patients without this complication. No relationship was observed between CIA-CS and IDH. IDH patients had a significantly higher UF rate; however, they did not differ compared to complication-free patients either in anthropometric or laboratory parameters. *Conclusions*: Patients with IDH were characterized by lower pre- and post-HD circumferential strain of the common carotid artery. The lower CCA-CS showed that impaired vascular reactivity is one of the most important risk factors for this complication’s occurrence.

## 1. Introduction

Intradialytic hypotension (IDH) is a common complication during hemodialysis (HD) sessions, occurring with a frequency of 4 to 39.9% [1,2,3,4,5,6]. The National Kidney Foundation’s Kidney Disease Outcomes Quality Initiative guidelines defined IDH as a decrease in systolic blood pressure (SBP) ≥ 20 mmHg during a HD session [7], but different definitions take into account a combination of blood pressure (BP) decline, its decline below a certain threshold, the occurrence of symptoms or the need for intervention [8]. However, Flythe et al.’s results suggest that nadir-based definitions (for pre-dialysis SBP <160 mmHg, nadir <90 mmHg and for pre-dialysis SBP ≥ 160 mmHg, nadir <100 mmHg) identify an association between IDH and all-cause mortality [9]. BP decline occurs due to a decrease in cardiac output and arterial tone, which are caused by insufficient response and/or impairment of compensatory mechanisms (cardiac, plasma refilling, passive venoconstriction, and an active increase in arterial tone) during HD [10]. Risk factors for IDH are both modifiable and non-modifiable, such as age >65 years, higher body mass index (BMI), female sex, cardiovascular disease, arrythmia, diabetes mellitus, vascular calcification (VC), hypoalbuminemia, hyperphosphatemia, anemia, higher ultrafiltration rate (UFR) and higher interdialytic weight gain (IDWG) [11,12,13,14,15,16,17,18,19]. An occurrence of IDH may depend on the composition of dialysate, e.g., sodium concentration and sodium profiling [20,21], concentration of calcium ions [22,23], bicarbonate [22] and temperature [21,24]. IDH is associated with unfavorable clinical outcomes, e.g., cardiovascular (CV) events, CV and all-cause mortality, vascular access thrombosis, loss of residual diuresis and development of dementia [1,2,4,9,25,26,27]. It seems that inadequate arterial reactivity is one of the mechanisms involved in the development of this complication. Carotid-femoral pulse wave velocity (PWV) is considered the ‘golden-standard’ measurement of arterial stiffness [28]. Two-dimensional speckle tracking is the software initially used to assess myocardial strain and strain rate [29,30]. However, numerous studies have demonstrated that it can also be useful in the assessment of arterial strain [31,32,33]. This ultrasonography non-doppler technique, based on the analysis of tissue motion, allows measurement of the circumferential strain (CS), radial strain (RS) and longitudinal strain (LS) of the carotid arterial wall [31,32,33]. It has been proven that, at least to some extent, these parameters (in particular CS) can be used to assess the local stiffness of vessels [32,33,34]. In our previous research, it was found that CS values decrease significantly in correlation with ultrafiltration volume during HD sessions [35]. This observation suggests that CS may be a parameter that shows the effectiveness of the cardiovascular response to a reduction of circulating plasma volume, and therefore correlate with the incidence of IDH. The aim of this study was to evaluate changes in the CS measurements of the common carotid artery (CCA) and common iliac artery (CIA) before and after an HD session, and to compare them to the incidence of IDH.

## 2. Materials and Methods

### 2.1. Study Population

Qualified for this cross-sectional study were 68 (26 F and 42 M) clinically stable, non-smoking, adult patients undergoing chronic 3 times weekly HD sessions in one dialysis center for at least 3 months.

### 2.2. Demographic and Clinical Data

Data regarding patients’ age, sex, etiology of ESRD, dialysis vintage, as well as comorbidities, were collected from patients’ medical histories and medical records.

The exclusion criteria were: a history of major CV complications within the preceding 3 months (acute or chronic atrial fibrillation, severe aortic stenosis, peripheral artery disease, recent myocardial infarction and cerebrovascular event, or decompensated congestive heart failure); morbid obesity (BMI > 40 kg/m^2^); signs of active infection on the day of the examination; active bleeding or a hemoglobin concentration below 8.0 mg/dL during the 3 months before a patient’s assessment; a bilateral arteriovenous (AV) fistula; wounds or dressings of the examined neck and groin areas; and lack of consent. 

### 2.3. HD Sessions

The patients were hemodialyzed for 3 to 5 h, three times per week, with low-flux synthetic, polysulphone membrane dialyzers (F series, Fresenius Medical Care AG, Bad Homburg, Germany) with Kt/V urea above 1.2. The blood flow was 200–380 mL/min, and the dialysate flow was 500–600 mL/min with bicarbonate dialysate (constant sodium concentrations of 138 mmol/L, calcium 1.25 mmol/L and potassium 2–4 mmol/L). The ultrafiltration (UF) rate was constant for individual patients during HD, and the dialysate temperature was 36.5 °C. Target dry weight was clinically evaluated based on the occurrence of peripheral oedema, pulmonary congestion and muscle cramps. 

### 2.4. Ultrasonography CS Measurements

Examinations were performed before and after the mid-week HD sessions. Before the examinations, patients were asked to rest at least 15 min in the supine position on their own dialysis beds. After the HD session (10 min at the latest), patients were examined in the supine position on the same dialysis bed. During the image acquisition of distal CCA and proximal CIA, patients were instructed to refrain from swallowing, and to hold their breath. All measurements were performed by the same radiologist, who was not aware of the HD courses. For all examinations, a GE Vivid I ultrasound machine with a linear probe 8 L RS in short-axis view and a frequency of 10 Mhz in the “carotid” preset, at an average of 43.7 frames per second, was used. The tracking quality was assessed and revised, as needed, during the examination. For each image acquisition, a concurrent ECG was recorded. For the purpose of analysis, at least two consecutive cardiac cycles were stored in the cine-loop format.

A total of 8 recorded sequences of dilation and contraction were analyzed per patient (2 sequences for each artery before and after the HD session). EchoPac software was used for the analysis (Figure 1). Detailed data of the analysis were described in another study [35].

### 2.5. Vital Signs

BP was measured on a dialysis bed with the patient in the supine position using an automated oscillometric device and optimal-sized cuff at the beginning (not earlier than 10 nor later than 5 min before HD began) and the end (not later than 10 min after HD ended) of HD, every 30 min of HD or in cases with IDH symptoms. BP was measured 5 min before ultrasonography measurements. IDH was defined for pre-dialysis as SBP < 160 mmHg, nadir < 90 mmHg, and for pre-dialysis as SBP ≥ 160 mmHg, nadir < 100 mmHg, during hemodialysis sessions while ultrasonography examinations were performed. Patients were divided into two groups: those with IDH episodes (IDH) and those without (nIDH). 

Each patient’s weight was measured before and after their HD session, and their height was measured before their HD session. BMI was calculated regarding body mass after HD as the mass in kilograms divided by the square of height in meters (kg/m^2^).

### 2.6. Biochemical Results 

Laboratory findings, including complete blood count (CBC) assessed during the past month and serum phosphate, total calcium and PTH levels from the 3 months preceding examination, were obtained from patients’ medical records. 

### 2.7. Statistical Analysis

Continuous data are presented as means with standard deviations (SD) or as medians with interquartile ranges, depending on the normality of the distribution, which was tested using Shapiro–Wilk’s test. Continuous data with normal distribution were compared using the unpaired Student’s *t*-test. Continuous non-normal distribution and ordinal data were compared using the Mann–Whitney U test. Nominal data are presented as numbers with percentages and were compared using the Chi2 test (if the number of cases in each subgroup exceeded 15) and the Chi2 test with Yates correction (if <5 cases were present in any subgroup). Correlations between continuous or ordinal variables were tested using Pearson or Spearman rank correlation tests, respectively. A logistic regression model with a stepwise backward feature selection was fitted to create a simple model predicting the occurrence of IDH that could be of use in clinical settings; to check its goodness of fit we performed ROC curve analysis. *p* values lower than 0.05 were considered statistically significant. All analyses were performed using Microsoft Excel (Microsoft, Redmond, Washington, USA) or Statistica 13.3 software (Dell, Rouind Rock, TX, USA).

## 3. Results

The characteristics of 66 analyzed patients are shown in Table 1. Two patients were excluded from the analysis due to poor visualization of the carotid artery due to numerous artifacts in the image. The causes of end stage kidney disease (ESKD) in examined patients were glomerular disease (*n* = 23), diabetic kidney disease (*n* = 13), hypertensive nephropathy (*n* = 9), polycystic kidney disease (*n* = 5), obstructive uropathy (*n* = 4), nephrocalcinosis (*n* = 1) and unknown (*n* = 11). Thirty-three patients were diagnosed with CV complications, e.g., coronary heart disease (n = 21), cerebrovascular disease (*n* = 5) and peripheral arterial disease (n = 17). Fifty-nine patients were hypertensive and were treated with angiotensin-convertase enzyme or sartans (*n* = 25), beta-blockers (*n* = 38), calcium blockers (*n* = 43) and other medications (*n* = 29). Thirty-one patients were treated with statins.

During HD sessions, 26 patients had an incidence of IDH (IDH group); nine of these were associated with symptoms; in two cases there were two IDH incidents during the session. In 14 cases, placing patients in the Trendelenburg position was applied as a rescue therapy, and nine patients were additionally infused with concentrated saline solution. IDH patients and those without this complication (nIDH) did not significantly differ in terms of age, dialysis vintage, systolic and diastolic BP before and after the HD session, BMI, or dry mass, but IDH patients had a significantly higher UF rate, as shown in Table 1. 

CCA-CS before and after HD differed significantly between IDH and nIDH groups. There were no significant differences in CIA-CS before or after HD sessions (Table 1). 

None of the evaluated biochemical results were significantly different in patients with or without IDH.

Multiple logistic regression analysis with forward stepwise variable selection showed that UF rate, height, dry mass BMI and CCA-CS post HD were factors associated with the occurrence of IDH. All model parameters and regression coefficients are presented in Table 2. The model had an AUC ROC of 0.8163 and allowed for identification of patients with IDH with 35.29% sensitivity and 91.84% specificity (Figure 2).

## 4. Discussion

In our study, IDH patients differed from those who did not experience this complication (with lower CCA-CS, both before and after HD) whereas the CS measured on CIA did not differ. To our knowledge, this is the first study investigating changes in CCA-CS and CIA-CS and the incidence of IDH. The observed phenomenon in IDH patients can be related to the processes taking place during HD sessions. It is postulated that the cause of hypotension is a reduced cardiac filling [36], primarily dependent on venous return (augmented mainly by plasma refilling) and accompanied by cardiac function, which determines cardiac output [37]. The content of the venous system constitutes approximately 60% of the total blood volume; it can be further divided from a functional point of view into theoretical stressed and unstressed parts [38]. The first generates a gradient between the venules and the right atrium, whereas the second is a reservoir. The main compartment of the unstressed volume is held in the splanchnic system [37]. It has been proven that during HD there is significant mobilization from this compartment [39], which plays a major role in cardiac filling. However, an autonomic system dysfunction and reduced sensitivity of baroreceptors in HD patients [40], combined with increased adenosine secretion (which further aggravates hypotension) [41], impairs the system and prevents splanchnic shift [39]. Venous return inadequacy may be further exacerbated by an increase in body temperature during HD [42] accompanied by dilatation of the skin vasculature [43,44] and intensified bradycardic and vasodilatory reflexes induced by sudden withdrawal of sympathetic activity and the predominance of parasympathetic tone [36]. Additionally, plasma refilling from interstitial and intracellular compartments ensures maintenance of blood pressure [45]. Moreover, the HD procedure itself causes tissue ischemia [46], which is associated with myocardial stunning [47] impairing cardiac function, a process even more severe in a significant percentage of patients (28.4%) with chronic heart failure. Cornelis et al. observed a fall in cardiac output (1.4 ± 1.5 L/min) during HD sessions [48]. As one of the independent factors, it leads to an increase in arterial stiffness (AS) and further aggravation of the IDH effect due to inadequate vessels reactivity [49,50]. The mechanisms responsible for increased AS in HD patients are still the subject of numerous studies; however, the most important factors identified include chronic volume overload, oxidant stress, chronic microinflammation, sympathetic overactivity, activation of the renin–angiotensin system [51,52,53] and VC [52,53,54].

Considering all the mentioned factors that have a satisfactory association between CCA-CS and VC [54], it appears that CS changes can simultaneously assess aspects related to refilling, venous return and cardiac output augmented by the relationship with AS. Moreover, significantly lower pre-dialytic CCA-CS in the IDH group indicated vascular stiffness as an important predisposing factor for IDH. Surprisingly, we did not find this difference in CIA-CS; perhaps it is related to the distance between the heart and examined vessels, but this observation requires further study.

Its small sample size from a single center and being a cross-sectional study with only a one-time measurement were the main limitations of our study. In addition, we used clinical assessment of the hydration status rather than bioimpedance. The definition of IDH we adopted has a significant clinical impact on dialysis patients; hence, the decision to use this definition. We undertook to eliminate or minimize deviations of modifiable factors affecting the frequency of IDH by selecting a group of patients using the same dialysate fluid composition (except potassium concentration), the same sodium concentration without profiling, at the same temperature; additionally, patients abstained from food ingestion during their HD sessions. Our results are in line with previous observations regarding the incidence of IDH in patients with higher UF rates [15,16,19].

## 5. Conclusions

Patients with IDH were characterized by lower pre- and post-HD circumferential strain of the common carotid artery. The lower CCA-CS showed that impaired vascular reactivity is one of the most important risk factors for this complication’s occurrence.

## Figures and Tables

**Figure 1 medicina-59-00102-f001:**
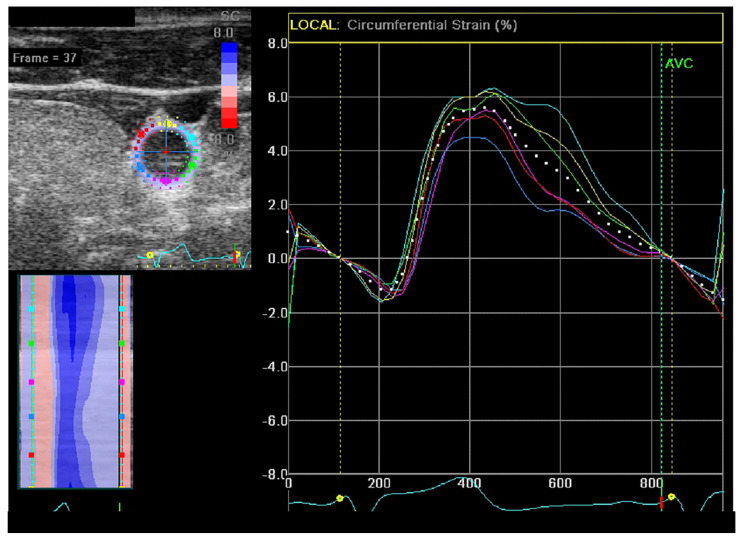
Carotid circumferential strain measurement analysis. Dotted graph indicates values for the whole circumference and every segment of the circumference is marked with different color.

**Figure 2 medicina-59-00102-f002:**
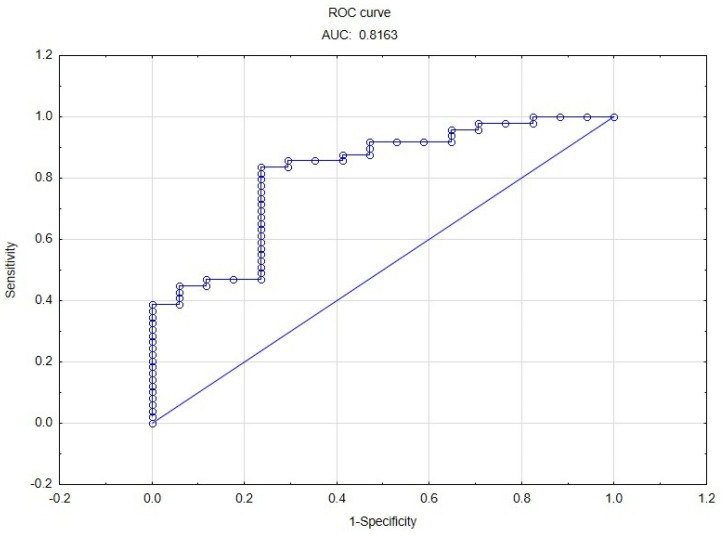
ROC curve of the model used to identify patients with IDH.

**Table 1 medicina-59-00102-t001:** Clinical (including laboratory) data and circumferential strain in patients without (nIDH) and with intradialytic hypotension (IDH).

	All Patients	No Intradialytic Hypotension (nIDH) (*n* = 40)	Intradialytic Hypotension (IDH) (*n* = 26)	*p*
Men/Women [%]	60.61/39.39	62.50/37.50	57.69/42.31	0.696
Age [years]	59.52, SD 15.28	56.63, SD 16.67	63.96, SD 11.82	0.056
HD vintage [months]	52 (9–86)	57 (9–88)	47.6, SD 30.69	0.582
Dry mass [kg]	69.02, SD 15.22	69.62, SD 15.49	68.10, SD 15.05	0.694
Pre-dialysis body mass [kg]	71.17, SD 15.60	71.61, SD 16.19	70.48, SD 14.94	0.777
Post-dialysis body mass [kg]	69.02, SD 15.60	69.62, SD 15.49	68.10, SD 14.94	0.694
Height [cm]	168.08, SD 8.62	169.15, SD 10.07	166.42, SD 5.52	0.212
BMI [kg/m^2^]	24.24, SD 4.06	24.13, SD 3.85	24.40, SD 4.42	0.789
IDWG [L]	2.15, SD 1.18	1.99, SD 1.19	2.39, SD 1.16	0.184
HD session duration [min]	240 (210–270)	239.88, SD 28.90	238.27, SD 30.69	0.830
UF rate [mL/h/kg]	7.91, SD 4.43	6.87, SD 3.45	9.51, SD 1.16	0.017
Dialyzer surface area [m^2^]	1.80 (1.80–2.10)	1.80 (1.80–2.10)	1.80 (1.80–2.10)	0.646
Dialysate potassium concentration [mmol/L]	3.00 (2.00–3.00)	3.00 (2.00–3.25)	3.00 (2.00–3.00)	0.651
CCA-CS [%]				
before HD	5.63, SD 2.26	6.28, SD 2.34	4.63, SD 1.74	0.003
after HD	4.28 (3.24–5.82)	5.00 (3.53–6.78)	3.57 (2.64–4.69)	0.010
difference	0.94, SD 1.72	1.00, SD 1.80	0.84, SD 1.56	0.705
CIA-CS [%]				
before HD	1.34 (0.69–2.08)	1.38 (0.70–2.11)	1.39, SD 1.05	0.808
after HD	1.13 (0.39–2.29)	1.45, SD 1.55	1.03 (0.69–1.83)	0.906
difference	0.07 (−0.77–0.79)	0.14 (−0.81–0.83)	−0.09 (−0.48–0.47)	0.694
Pre-HD HR [beats/min]	67.02, SD 13.37	69.79, SD 11.89	59.25 (51.00–67.00)	0.014
Pre-HD SBP [mmHg]	131 (115–152)	131.80, SD 20.23	138.62, SD 31.90	0.291
Pre-HD DBP [mmHg]	75.50, SD 13.33	75.82, SD 11.60	70.50 (66.00–82.00)	0.362
Pre-HD MAP [mmHg]	95.16, SD 15.42	94.48, SD 12.50	96.21, SD 19.30	0.661
Post-HD HR [beats/min]	74 (65–87)	77.49, SD 13.29	66.50 (53.00–89.00)	0.096
Post-HD SBP [mmHg]	120 (107–137)	124.68, SD 17.77	109 (102–132)	0.128
Post-HD DBP [mmHg]	72.64, SD 11.50	74.38, SD 11.17	69.96, SD 11.71	0.129
Post-HD MAP [mmHg]	89.38, SD 13.47	91.14, SD 12.44	81.33 (75.00–96.00)	0.082
HGB [g/dL]	10.65, SD 1.45	10.40, SD 1.21	11.03, SD 1.71	0.083
HCT [%]	31.83, SD 4.08	31.16, SD 3.33	32.85, SD 1.71	0.099
Ca [mmol/L]	2.20 (2.11–2.32)	2.23, SD 0.15	2.14 (2.09–2.33)	0.309
P [mmol/L]	1.83, SD 0.52	1.90, SD 0.55	1.72, SD 0.46	0.172
PTH [pg/mL]	300.90 (116.45–523.80)	374.83, SD 249.32	213.90 (102.73–435.90)	0.243

HD: hemodialysis; BMI: body mass index; IDWG: interdialytic weight gain; UF: ultrafiltration; CCA-CS: circumferential strain of common carotid artery; CIA-CS circumferential strain of common iliac artery; HR: heart rate; SBP: systolic blood pressure; DBP: diastolic blood pressure; MAP: mean arterial pressure; HGB: hemoglobin concentration; HCT: hematocrit; Ca: calcium; P: phosphorus; PTH: parathormone. Nominal variables are presented as absolute and relative numbers and continuous data are presented as means with standard deviations or as medians with the values of upper and lower quartiles.

**Table 2 medicina-59-00102-t002:** Parameters in the model used to identify patients with IDH.

Parameter	β	SE	OR (95% CI)	*p*
Intercept	−11.43	8.26		0.167
UF rate [mL/h/kg]	−0.19	0.09	0.83 (0.70–0.98)	0.028
Height [cm]	0.10	3.57	1.11 (1.00–1.23)	0.059
BMI [kg/m^2^]	−0.16	0.10	0.85 (0.70–1.03)	0.089
CCA-CS after HD [%]	0.33	0.21	1.39 (0.92–2.10)	0.116

Β: regression coefficient; SE: standard error; OR: odds ratio; CI: confidence interval; UF: ultrafiltration; BMI: body mass index; CCA-CS: circumferential strain of common carotid artery.

## Data Availability

The data presented in this study are available on request from the corresponding author. The data are not publicly available due to privacy restrictions.

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
