# Peer review of "Circumferential Strain as a Marker of Vessel Reactivity in Patients with Intradialytic Hypotension"

_medicina, 2023, doi:10.3390/medicina59010102_

Round 1

Reviewer 1 Report

Goździk and co-workers propose a study in which they evaluate the changes

in the CS measurements of the common carotid artery (CCA) and common iliac

artery (CIA) before and after HD session and relate them to the incidence of IDH.

 They conclude that the patients with IDH were characterized by lower pre and post- HD circumferential strain of the common carotid artery and that the lower CCA-CS impaired vascular reactivity is one of the most important risk factors of this complication

occurrence.

Generally the technical part of this work seems to be well conducted and performed. The procedures and techniques used are standard and appear appropriate. 

As the authors themselves point out, the only limitation of the proposed study is  the small sample size from a single center and the cross-sectional study with only one-

time measurement, however I find the study interesting

Author Response

Thank you very much for your review and positive opinion.

Reviewer 2 Report

The authors analyzed the value of sonography-obtained circumferential strain of carotid artery as a marker of vessel reactivity in patients with and without intradialytic hypotension (IHD). It is very important clinical issue, as IHD incidents is a confirmed risk factor for cardiovascular complications in this high-comorbidity population.

However, several comments should be addressed to improve the merit value of this work.

Main comments:

1.       The statistics should be improved. In a whole study group, please provide the univariate logistic analysis of all parameters presented in Table 1 with p values < 0.1, and then multivariate analysis for the occurrence of IHD as dependent variable and the abovementioned potential explanatory variables (note: please use only Hb values, not Hb and Ht in this analysis).

2.       The amount of residual diuresis should be reported in Table 1 and included in further analyses.

3.       If available, the hydration status parameters obtained using bioimpedance analysis performed before and after the analyzed HD sessions might be further improvement of the presented analysis.

4.       To enhance the educational value of this publication, it would be useful to add the illustration/photography to visualize how the arterial circumferential strain was measured in carotid arteries.

5.       Please write in detail how were patients classified as having IHD or not: based on single dialysis session, 3 sessions during the examinations week or the longer timespan?

6.       In the Discussion, the authors should more clearly state what is the place of CCA-CS measurement in the armamentarium of diagnostics methods in HD patients and how to use this parameter for more accurate patients care.

7.       The careful English editing is necessary.

Minors:

1.       line 158-162: several results are given twice: in the text and in Table 1: the numeric values in the text should be omitted.

2.       Table 1: the most of presented parameters should be presented with only one cipher after the comma or even without decimals (HR, HD duration session etc.).

3.       CCA-CS values are presented in different ways (means or medians) depending on their normality analysis. However, as these values are finally used for the difference calculation, please present them in both time-points as medians with IQRs to calculate the difference using non-parametric test (you may present pre-HD values in both ways for better clarity).

Author Response

The statistics should be improved. In a whole study group, please provide the univariate logistic analysis of all parameters presented in Table 1 with p values < 0.1, and then multivariate analysis for the occurrence of IHD as dependent variable and the abovementioned potential explanatory variables (note: please use only Hb values, not Hb and Ht in this analysis).

Thank you very much for your suggestion. The analysis has been added to manuscript.

  1. The amount of residual diuresis should be reported in Table 1 and included in further analyses.

      Thank you very much for bringing this to our attention. Unfortunately, we did not collect this data at the time of the ultrasound examination so we are not able to add and analyze it at this study.

  1. If available, the hydration status parameters obtained using bioimpedance analysis performed before and after the analyzed HD sessions might be further improvement of the presented analysis.

      Thank you very much for bringing this to our attention. In the study, we decided to only clinically assess the patients’ hydration status. We did not perform an assessment using bioimpedance. We added this information as a limitation of the study.

  1. To enhance the educational value of this publication, it would be useful to add the illustration/photography to visualize how the arterial circumferential strain was measured in carotid arteries.

      As recommended, the graphic of the measurement has been added.

  1. Please write in detail how were patients classified as having IHD or not: based on single dialysis session, 3 sessions during the examinations week or the longer timespan?

      As suggested, information was added that the diagnosis of IDH concerned this particular hemodialysis session during which ultrasound was performed. In the “2.4 Ultrasonography CS measurements” section we have already placed information that the ultrasonography were performed during the mid-week dialysis session.

  1. In the Discussion, the authors should more clearly state what is the place of CCA-CS measurement in the armamentarium of diagnostics methods in HD patients and how to use this parameter for more accurate patients care.

      We did not specify the place of CCA-CS in HD patients care because it was not a purpose of our study. We only focused on CCA-CS as a marker of vessel reactivity. Since the data were collected from only one dialysis session, and no preventive action was taken on the basis of CCA-CS before the procedure, we are afraid that drawing conclusions about the CCA-CS place in patient care would not be supported by the results, but only by our assumptions. The next study we are preparing will deal with the issue of modification of therapy and an attempt to prevent IDH on the basis of CCA-CS.

  1. The careful English editing is necessary.

Thank you very much for this suggestion. Minor changes were performed.

Minors:

  1. line 158-162: several results are given twice: in the text and in Table 1: the numeric values in the text should be omitted.

Thank you, we have made a correction.

  1. Table 1: the most of presented parameters should be presented with only one cipher after the comma or even without decimals (HR, HD duration session etc.).

Thank you, we have made a correction.

  1. CCA-CS values are presented in different ways (means or medians) depending on their normality analysis. However, as these values are finally used for the difference calculation, please present them in both time-points as medians with IQRs to calculate the difference using non-parametric test (you may present pre-HD values in both ways for better clarity).

Thank you, we have made a correction.

Round 2

Reviewer 2 Report

The authors markedly improved the scientific and educational value of the manuscript.